# Infection of Mammary Glands of Small Mammals in Eastern North America by Helminths

**DOI:** 10.3390/ani11113207

**Published:** 2021-11-10

**Authors:** David Bruce Conn, Cary A. Hefty, Sarah Cross Owen

**Affiliations:** 1One Health Center, Department of Biology, Berry College, Mount Berry, GA 30149, USA; carysanders423@gmail.com; 2Department of Invertebrate Zoology, Museum of Comparative Zoology, Harvard University, Cambridge, MA 02138, USA; 3West Virginia University Extension Service, West Virginia University, Morgantown, WV 26506, USA; Sarah.Owen@mail.wvu.edu

**Keywords:** cotton rat, helminth, maternal transmission, nematode, parasitic, Rodentia, Soricimorpha, transmammary, vertical transmission

## Abstract

**Simple Summary:**

For several decades, it has been known that many species of parasitic worms can infect the mammary glands of their hosts, and that some of these can be transmitted through the mammary glands from mother to nursing young. Such infections have been studied widely in many species of wild and domesticated animals, including livestock as well as cats and dogs, and also in humans. Despite this, field studies specifically designed to determine whether mammary infections occur in wild mammals sampled from their natural environment have not been performed. We explored this by conducting a survey of wild small mammals from natural sites in the states of New York, Tennessee, and Georgia in the eastern United States. We examined 53 wild hosts, including four species of rodents and one shrew, using a new method of removing and mounting whole mammary glands for microscopical study. Many intestinal parasites were found, but worms occurred in the mammary glands of only one species; four cotton rats from Georgia had roundworm larvae in their mammary glands. This is the first general survey of wild mammals to include mammary examination. Based on our positive results, we propose inclusion of mammary examination in all future surveys.

**Abstract:**

To determine whether small mammals living in natural settings harbor helminth infections in their mammary glands, we conducted a survey of helminths infecting rodents and soricimorphs in three widespread locations in the eastern United States: states of New York, Tennessee, and Georgia. We examined all the primary organs in all hosts, and identified all helminths. We also excised the complete mammary glands within their subcutaneous fat pads, then stained and mounted each whole mammary gland set for microscopical examination. A total of 53 individual hosts were examined, including 32 *Peromyscus* spp., 11 *Mus musculus*, 5 *Sigmodon hispidus*, 4 *Clethrionomys gapperi*, and 1 *Blarina carolinensis*. Helminths collected included *Heligmosomoides* sp., *Hymenolepis*
*diminuta*, *Hymenolepis nana*, *Pterygodermatites peromysci*, *Schistosomatium douthitti*, *Syphacia obvelata*, *Syphacia sigmodontis*, and *Trichostrongylus sigmodontis*. Four *S. hispidus* were infected by *T. sigmodontis* in the small intestine; in all four, we also found nematode larvae in lactiferous duct lumen and lactogenic tissue of the mammary glands. We were unable to identify the species of nematode larvae, but the co-occurrence with *T. sigmodontis* in all cases may suggest an association. Future studies should seek to identify such larvae using molecular and other methods, and to determine the role of these mammary nematode larvae in the life cycle of the identified species. No other host species harbored helminths in the mammary glands. Overall, our results suggest that mammary infections in wild small mammals are not common, but warrant inclusion in future surveys.

## 1. Introduction

For several decades, it has been known that some helminth parasites of mammals are able to infect the mammary glands, and in some cases, establish infections in the nursing young through transmammary transmission. Several reviews have been published on this subject over many years [1,2,3,4,5]. The earliest work on transmammary transmission was that of Lyon and associates, who showed that fur seals pass infective larvae of the nematode, *Uncinaria lucasi*, from mother to young through the milk [6]. However, most of our knowledge of mammary infections and transmammary transmission is from studies on domesticated large mammals [6], companion animals [7], experimental laboratory rodents [8,9,10], or clinical reports from humans [11,12,13]. At least one group of heligmosomatid nematodes, of the genera *Mammaniduloides*, *Mammanidula*, and *Mammalongistriata*, live as adults in the mammary gland of some small rodents in Asia [14]. However, general surveys of helminths from wild, naturally infected hosts typically do not include examination of the mammary glands, so little is known about how frequently mammary infections occur in natural conditions. Learning more about this is critical to understanding the full dynamics of parasite–host systems in wild populations. To explore this, we undertook a field survey of wild small rodent and soricimorph mammals in three widespread locations in eastern North America.

Conn [5,8,9] developed a method to remove and examine whole mammary glands from small laboratory rodents so that localization within or around mammary tissue could be ascertained. This technique has never been applied to naturally infected wild small mammals, so we incorporated it into our studies so that we could determine whether mammary infections occur, and could simultaneously compare the incidence of these with other helminthic infections present in those hosts.

## 2. Materials and Methods

### 2.1. Collection and Examination of Hosts

Hosts examined in this study were collected from three disjunct regions in the eastern United States, chosen to cover a wide range of biogeographical locations throughout the continent: (1) northern New York state (St. Lawrence County) in the St. Lawrence River Valley at the edge of the Adirondack Mountains; (2) south-central Tennessee (Franklin and Marion Counties) atop the Cumberland Plateau (plus 1 mouse from Washington County in eastern Tennessee); (3) northwest Georgia (Floyd County) on the Berry College campus in the Ridge-and-Valley region, at the northern edge of the Piedmont topographical region. Trapping occurred at various times of the year.

All hosts were collected in Sherman, Havahart, or Tin Cat live traps baited with peanut butter, then transported to the laboratory where they were euthanized by isoflurane vapor inhalation followed by cervical dislocation. Most were necropsied immediately after euthanasia but some were placed in plastic bags and frozen prior to necropsy. Each mammal was weighed, measured from snout to anus, then examined in detail for helminths. Necropsy began with performing shallow ventral incisions through the skin but without penetrating the musculature of the body wall. The skin was carefully removed to allow removal of the subcutaneous fat pads containing the mammary tissue as described previously [1,4,5]. All 4 of the fat pads, 2 anterior and 2 posterior, each containing mammary glands, were removed and fixed immediately in hot 10% neutral buffered formalin while being pressed gently between two glass plates.

Following fat pad/mammary tissue removal, the body cavity was opened and all thoracic, abdominal, and pelvic viscera were removed and examined thoroughly for helminths. All parts were bathed frequently with physiological saline during all stages of examination. The diaphragm was removed and examined fresh with a compound light microscope while pressed gently between glass slides then fixed with 10% neutral buffered formalin or while still between the slides.

### 2.2. Processing and Examination of Helminths and Mammary Tissue

All helminths were removed, rinsed quickly in physiological saline, and fixed by immersion in hot neutral buffered formalin or ethanol–formalin–acetic acid for at least 24 h. Fixed cestodes, trematodes, mammary fat pads, and diaphragm pieces were rinsed in water, stained in Semichon’s or Ward’s acetocarmine, dehydrated in an ascending series of aqueous ethanol, cleared in xylene or methyl salicylate, and mounted whole on glass microscope slides in gum damar. Fixed nematodes were rinsed in water, dehydrated in an ascending series of aqueous ethanol, cleared in Langeron’s lactophenol, then double mounted in glycerine jelly between two glass coverslips that were then sealed onto microscope slides with gum damar.

All material was examined with dissecting microscopes and with compound light microscopes using brightfield and differential interference contrast optics, and photographed with digital cameras integrated with the microscopes. Voucher specimens of mammary glands and some helminths were deposited in the Museum of Comparative Zoology at Harvard University. The handling of all animals at all stages of this work was approved by the Institutional Animal Care and Use Committee of the respective institutions.

## 3. Results

A total of 53 individual hosts belonging to 5 mammal species (4 Rodentia and 1 Soricimorpha) were examined. These harbored a total of eight species of adult helminth parasites that were identified to species or genus, plus one unidentified nematode larva (Table 1).

Fourteen of the fifty-three (26.4%) mammals examined harbored at least one helminth species. Six of the thirty-two *Peromyscus leucopus* (18.8%) were infected; three of the eleven *Mus musculus* (27.3%) were infected; all five (100%) of the five *Sigmodon hispidus* were infected; none (0%) of the four *Clethrionomys gapperi* and *Blarina carolinensis* were infected.

The only mammary glands that harbored helminths were those of four *S. hispidus*. In these hosts, nematode larvae occurred within the mammary gland tissues, including the lactiferous ducts of the two infected males (Figure 1A) and the two infected females (Figure 1B), as well as in the lactogenic alveolar areas of the two female hosts (Figure 1C).

All four of the rats that were infected by larvae in their mammary glands were also infected with large numbers of adult male and female *Trichostrongylus sigmodontis* in their small intestine (mean = 42 worms; range, 25–53 worms). Only one *S. hispidus*, a female, lacked mammary larvae, and also lacked *T. sigmodontis* adults. This female *S. hispidus* was infected with five adult *Hymenolepis diminuta* in the small intestine.

## 4. Discussion

This report is the first study specifically looking for the occurrence of helminths in the mammary glands of wild hosts collected from their natural environment. It is also the first to use this new method of removing and mounting whole mammary gland sets from each host in a field survey. The method has been used for laboratory studies [5,8,9], where it was effective in demonstrating mammary infection and transmammary transmission of juvenile cestodes (metacestodes) in mice that were experimentally infected by oral feeding and intraperitoneal injection in an artificial setting. In the present report, we described four naturally occurring infections of *Sigmodon hispidus* mammary glands with nematode larvae. This host species has been studied extensively, but the lack of reports of mammary helminths may relate to the fact that those studies did not mention examination of the mammary glands [15,16,17,18,19,20,21]. Likewise, the examination of mammary glands has not been mentioned in other surveys of rodent helminths in the region [22,23].

The use of whole mounts in this study did not allow for the determination of a specific pathology due to the presence of helminths in the mammary glands. Future studies might explore histopathological changes in the mammary tissue as reported in experimentally infected laboratory mice in earlier studies [5,9]. This will require histological methodology not employed in the present study. Moreover, the stages of pregnancy and lactation may affect the occurrence of helminths in the mammary glands, as has been reported for other hosts [2,3]. Future studies might explore this by applying standardized methods for determining the stage of pregnancy and/or lactation in the affected host individuals.

The sample sizes for this study were modest, but the finding of mammary infections in 4/53 (>7.5%) of all small mammal hosts examined, and 4/5 (80%) of one host species, suggests that mammary infections are not rare in nature and should be considered in future surveys. Certainly, this whole-mount method of examining host mammary glands is effective in surveying small mammals such as the rodents and shrews reported here. Other mammals that might be studied in this way would be moles (Talpidae) and some other groups of small mammals. However, the method as described here is suitable only for very small mammals, and the *S. hispidus* studied here is probably at the larger end of the range for mammals whose mammary gland fat pads can be mounted completely on microscope slides. However, based on the success of this present study, we propose that examination of mammary glands should become part of the protocol for surveys of all wild mammalian hosts, given the fact that mammary infections, often including transmammary transmission, are so common among large domesticated mammals [2,14]. For mammals that are substantially larger than rodents or shrews, other methods should be explored. For example, we suggest removing mammary tissue from larger hosts to process through a Baermann funnel apparatus, which is used to isolate migrating nematode larvae from other organs. Other methods of extracting parasites from tissue might also be considered, such as various tissue digestion techniques. The method we describe here is easy, and quick to perform and to teach, so it is very suitable for studies of small mammals. Indeed, we may never fully understand the epizootiology of natural helminth infections if we continue to disregard mammary infections.

Our experimental approach in this study did not permit identification of the mammary gland larvae. Future studies should make this a priority by using molecular methods that are generally available and commonly practiced. The fact that all four of the *S. hispidus* individuals that harbored nematode larvae were also heavily infected with *Trichostrongylus sigmodontis*, while the *S. hispidus* with no mammary larvae harbored no *T. sigmodontis*, invites speculation that the larvae were in fact *T. sigmodontis*. However, such speculation would not be prudent until further research is conducted. The only research performed in the past on the life cycle of *T. sigmodontis* did not consider transmammary transmission, but only ingestion of larvae as a route to infection [24]. Moreover, there are other common nematode parasites of rats that have been shown experimentally to engage in transmammary transmission in laboratory studies, including *Strongyloides ratti* [25] and *Strongyloides venezuelensis* [26]. Others such as *Nippostrongylus brasiliensis* can have tissue-migratory larvae in *S. hispidus* [27]. None of these species were found in the present study, but it is possible that they were present in the environment and the mammary infections reported here were from penetration from the soil. The occurrence of larvae in the present study specifically localized within the lactiferous ducts and lactogenic alveolar areas argues against random subcutaneous migration, but further work is needed.

Because of the novel approach used in this study, we did not prepare any of the nematodes for molecular analysis. Furthermore, the treatment of our mammary glands by fixation in formalin prevents retrospective examination by DNA sequencing, PCR, or other molecular approaches. However, we suggest that, in the future, these approaches should be incorporated. The simplest solution would be to remove some of the mammary glands from each host for microscopical identification, while fixing the remaining mammary tissue in ethanol, preferably after extracting the nematode larvae. After the species is determined, it will still be necessary to conduct experimental infections in the laboratory to confirm transmammary transmission, but the less demanding goal of establishing the nature and prevalence of mammary infections in field samples must be a priority. This is especially true for nematodes, which are the helminths most commonly transmitted through the mammary glands [28]. However, since it has been shown that proliferative metacestodes can occur in mammary glands in experimentally infected rodents [5,8,9], it will also be important in the future to consider this as a possibility in other aberrant metacestodes, such as those we have reported from some locations in Europe [29,30].

## 5. Conclusions

Results of this survey showed patterns of parasite community structure and prevalence that are comparable to other biotic surveys of similar extent, with several typical helminth species being recorded. However, it is the first time that there was a finding of parasitic nematodes in the mammary glands for such a study conducted on wild host populations. This shows that using this unique method of whole mammary gland extraction and whole mount processing is an effective approach for small mammals and should become a standard part of rodent and soricimorph surveys worldwide, as the technique is simple to perform and to teach. Other methods of mammary gland examination will be required for larger mammals, but should be developed and used, because mammary infection is likely to be more common than past research has suggested.

## Figures and Tables

**Figure 1 animals-11-03207-f001:**
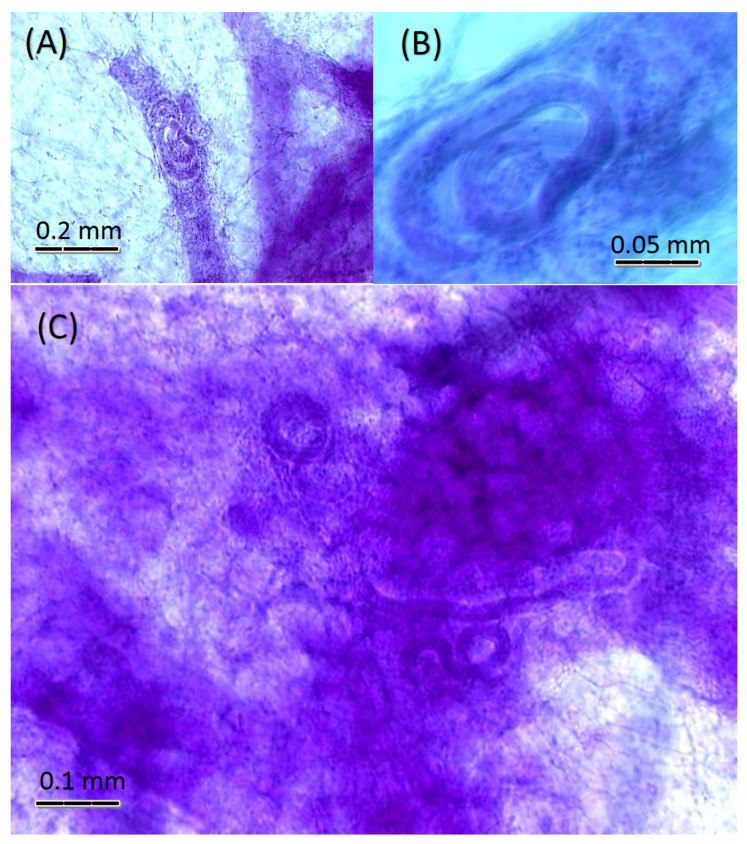
Whole mounts of acetocarmine-stained mammary glands within subcutaneous fat pads of Sigmodon hispidis, showing infections by nematode larvae: (**A**) from male host with larva inside lactiferous duct. (**B**) From female host showing larva inside lactiferous duct. (**C**) From female host showing several larvae among lactogenic alveoli and connective tissues of fat pad.

**Table 1 animals-11-03207-t001:** Hosts examined from specified states with data on parasites recovered during necropsies.

Host Species	Collection Site	Number of Hosts Examined	Parasite Identity (Number of Hosts Infected; Prevalence as % Infected)
*Blarina carolinensis* Southern short-tailed shrew	Georgia	1	None present (0; 0%)
*Clethrionomys gapperi* Southern red-backed vole	New York	4	None present (0; 0%)
*Mus musculus* House mouse	Georgia	11	*Pterygodermatites coloradensis* (1; 9%) *Syphacia obvelata* (2; 18%)
*Peromyscus leucopus* White-footed mouse	New York	5	*Hymenolepis nana* (2; 40%) *Schistosomatium douthitti* (1; 20%)
*Peromyscus leucopus*	Tennessee	25	*Pterygodermatites coloradensis* (3; 12%)
*Peromyscus leucopus*	Georgia	2	*Heligmosomoides* sp. (1; 50%)
*Sigmodon hispidus* Hispid cotton rat	Georgia	5	*Hymenolepis diminuta* (1; 20%) *Syphacia sigmodontis* (1; 20%) *Trichostrongylus sigmodontis* (4; 80%) Mammary nematode larvae (4; 80%)
Total (5 host species)	3 states	53	8 species of adult; 1 species of larva (14; 26.4%)

## Data Availability

Voucher specimens of mammary glands and some helminths were deposited in the Museum of Comparative Zoology at Harvard University.

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
