# Peer review of "Infection of Mammary Glands of Small Mammals in Eastern North America by Helminths"

_animals, 2021, doi:10.3390/ani11113207_

Round 1

Reviewer 1 Report

The manuscript "Helminths of Small Mammals in Eastern North America with 2 Emphasis on Infections of Mammary Glands" is an interesting study that highlighted the presence of helminths in mammary glands.

In general, the study was well written and discussed. 

I would like to know what is the economical or environmental importance of the mammary gland infection by helminths.

One point that I have a concern about is the euthanasia method. In the USA the legislation or veterinary councils do not recommend the use of a high dose of central nervous system anesthetics (inhalation or intravenous) instead of carbon dioxide? Can asphyxia cause pain in mammals?

I suggest including the popular name of species in table 1.

The word infection in veterinary medicine means a disease: the animals examined were illness? 

Is there no correlation between mammary gland presence of parasites and intestinal helminths?

These animals can live together with these parasites without any health problems? 

Can the parasites benefit these mammals against other diseases? Do these animals produce IgE or mucosal IgA against these parasites?

Reviewer 2 Report

This is a very nice, and well written manuscript investigating inter mammary infection with parasites. This is an unusual angle to take, so is very nice to see and to read. It would have been nice to see more samples included, but even 1000 samples wouldn’t detract from the interest and importance of the results so I have no issues here. The manuscript is only short, but appropriate for the data collected. 

I have a few very minor comments, but overall this is a lovely manuscript, and I commend and thank the authors for making a great manuscript and a reviewers life easy!

Line 34, comma after 4

Line 46- comma after cases

Line 48- the last several decades sounds a bit strange. Maybe consider rewording, but up to the authors

Line 79- did freezing affect the results at all?

Line 83- I would probably say previously rather than earlier?

Table 1- could you add in the name which these are commonly known by?

Also in table 1- could you add in the % infected in each species here as well?

Line 140- I would delete ‘for a’

Line 142- first to describe the use of this … (Reword)

Line 194- comma after that and future may aid reading here

But overall, very minor comments and a very nice manuscript
